# Room temperature electrically pumped topological insulator lasers

Jae-Hyuck Choi[1], William E. Hayenga[1,2], Yuzhou G. N. Liu [1], Midya Parto [2], Babak Bahari[1], Demetrios N. Christodoulides [2] & Mercedeh Khajavikhan [1,2 ✉]

Topological insulator lasers (TILs) are a recently introduced family of lasing arrays in which phase locking is achieved through synthetic gauge fields. These single frequency light source arrays operate in the spatially extended edge modes of topologically non-trivial optical lattices. Because of the inherent robustness of topological modes against perturbations and defects, such topological insulator lasers tend to demonstrate higher slope efficiencies as compared to their topologically trivial counterparts. So far, magnetic and non-magnetic optically pumped topological laser arrays as well as electrically pumped TILs that are operating at cryogenic temperatures have been demonstrated. Here we present the first room temperature and electrically pumped topological insulator laser. This laser array, using a structure that mimics the quantum spin Hall effect for photons, generates light at telecom wavelengths and exhibits single frequency emission. Our work is expected to lead to further developments in laser science and technology, while opening up new possibilities in topological photonics.

[1] Ming Hsieh Department of Electrical and Computer Engineering, University of Southern California, Los Angeles, CA, USA. [2] CREOL, The College of Optics and Photonics, University of Central Florida, Orlando, FL, USA. ✉email: khajavik@usc.edu

I n condensed matter physics, topological insulators (TIs) represent a new class of materials with insulating bulks and conducting symmetry-protected surface states[1–9]. The remarkable robustness of these surface currents against local defects and perturbation has led to the application of these materials in quantum transport, spintronic devices, and new transistors, to name a few[4–9]. While the concept of topological protection was originally conceived for fermionic systems, recent advances have led to designing lattices capable of mimicking analogous responses in the electromagnetic domain[10–19]. This is primarily accomplished by realizing artificial gauge fields that emulate the effect of external magnetic fields on light particles through geometric design or modulation. In this respect, photonics has made it possible to study some of the intriguing aspects of topological physics by providing access to synthetic dimensions[20–23] and higher-order topological insulators[24–26]. Also, photonics allows the studying of topological behaviors arising due to non-Hermiticity (primarily gain) and nonlinearity that have no immediate counterparts in condensed matter[27–33]. In return, topological physics has inspired novel unique light transport schemes that may have important ramifications in integrated photonics[34,35], quantum optics[36,37], and laser science[38–50].

Topological insulator lasers (TILs) refer to two-dimensional arrays of emitters that oscillate at the topological edge modes. In photonics, the topological lattice is an artificial structure, designed to generate a synthetic gauge field for photons through direction-dependent phase accumulations. The edge modes propagating at the boundary of these lattices feature transport, thus uniformly engaging all the active elements at the perimeter of the array. The manifestation of this type of mode indicates that the active peripheral entities are phase-locked and the system has reached an optimum usage of the pump power[40]. Furthermore, the robustness of the edge modes to certain classes of perturbations introduces additional benefits by precluding the formation of spurious defect states. These undesired modes tend to compromise the performance of the device by siphoning energy from the main mode. Finally, the additional spectral modes of the cavity can be suppressed by controlling the topological bandgap through adjusting the array parameters. This latter feature is particularly important in semiconductor gain systems with intrinsically broad lineshapes in which single-frequency lasing poses a challenge.

So far topological lasing has been demonstrated in optically pumped two-dimensional topological insulator lasers based on active lattices featuring quantum Hall, quantum spin Hall, or valley Hall effect in the absence and presence of external magnetic fields[38–45]. In addition, lasing in the edge or zero modes has been achieved in one-dimensional SSH arrays, through PT-symmetric[46–49] or other selective pumping schemes[50]. Despite the rapid developments in the optically pumped topological insulator lasers, devices based on electrical injection are still in their infancy. What makes this transition a challenge is designing a topological structure that allows for both efficient carrier injection and large mode confinement. To this end, recently, an electrically pumped THz quantum cascade topological insulator laser was demonstrated[44] in which the edge mode is formed at the boundary of two valley Hall photonic crystals possessing valley Chern numbers of ±1/2. However, in order to reach lasing, this array was cooled to a cryogenic temperature of 9 K.

Here, we report the first room temperature, electrically pumped topological insulator laser that operates at telecom wavelengths. Our non-magnetic topological insulator array imitates the quantum spin Hall effect for photons through a periodic array of resonators coupled through an aperiodic set of auxiliary link structures. When the resonators at the perimeter of the array are electrically pumped, a uniform and coherent edge mode is excited. We further demonstrate that the topological property of the system gives rise to single-frequency lasing, despite the presence of multiple modes when the array's peripheral elements are locally excited. Our work shows for the first time the feasibility of realizing electrically pumped room temperature topological laser arrays that upon further developments can drive technological needs in related areas.

## Results

**Design and fabrication.** A schematic of our electrically pumped topological insulator laser is depicted in Fig. 1a. The array is composed of a $10 \times 10$ network of microring resonators coupled via a set of anti-resonant link objects, all fabricated on a III–V semiconductor wafer. Figure 1b displays the SEM image of the topological array in an intermediate fabrication stage. The structure implements the quantum spin Hall Hamiltonian for photons by modulating the relative position of the links in each row (see Fig. 1c)[12,15,40]. In order to promote edge mode lasing, the gain is only provided to the peripheral elements through the incorporation of metal electrodes, while the rest of the array is left unpumped to prevent spurious lasing in the bulk. The wafer as shown in Fig. 1d is composed of lattice-matched epitaxially grown layers of $In_xGa_{1-x}As_yP_{1-y}$ on an undoped InP substrate (see Supplementary Note 1 for details about wafer structure). The design of the wafer structure and the geometry of individual ring resonators are co-optimized in order to efficiently funnel electron and hole carriers into the active region under electrical pumping while at the same time allowing the structure to support a confined optical mode that adequately overlaps with the multiple quantum wells (MQWs). Figure 1e shows the structure of a single resonator at the boundary of the lattice.

To demonstrate lasing by electrical pumping at room temperature, the radii of the ring resonators ($R$) and their widths ($w$) are chosen to be 15 μm and 1.4 μm, respectively. The cross-section of the resonators is designed to promote lasing in the fundamental transverse electric mode ($TE_0$). (see Supplementary Note 2 for details about electromagnetic simulations). The links are of the same widths ($w$), but their radii of curvature ($R_L$) are 2.25 μm and the lengths of their straight sections ($L_L$) are 3 μm. Here the links are intentionally designed with a small radius of curvature in order to prevent their direct contribution through standalone lasing. The gap size between the ring resonators and the off-resonant links ($s$) is designed to be 200 nm. This allows a frequency splitting of ~31 GHz (0.245 nm in wavelength) between two neighboring resonators (see Supplementary Note 3 for deriving the coupling coefficient through measuring frequency splitting). The coupling strength dictates the topological bandgap of the structure, hence effectively determining the longitudinal modal content of the cavity. A stronger coupling also indicates that the edge mode can withstand larger defects, perturbations, and detunings[12,15,40]. In order to emulate a synthetic gauge field, throughout the array, the position of the links is judiciously vertically shifted from one row to another. Depending on the offset of the link resonators, the photon acquires an asymmetric phase of $\pm 2\pi\alpha$, where $\alpha$ is given as a function of the position shift $\Delta x$, such that $\alpha = 2n_{\text{eff}}\Delta x/\lambda$[15,40]. Here $\alpha$ is designed to be 0.25 corresponding to a $\Delta x$ of 60 nm. For a given $\Delta x$, the photonic dynamics are equivalent to having $a$ quanta of synthetic magnetic flux penetrating each plaquette. This results in two topologically nontrivial edge states at the boundary of the lattice[12,15,40].

The laser arrays are fabricated on a molecular beam epitaxy (MBE) grown InGaAsP/InP semiconductor wafer by using several stages of lithography and etching, as well as deposition of various materials (metals and dielectrics). The gain region consists of an undoped 10-multiple quantum well film with a thickness of 320 nm, sandwiched between the $n$-doped and $p$-

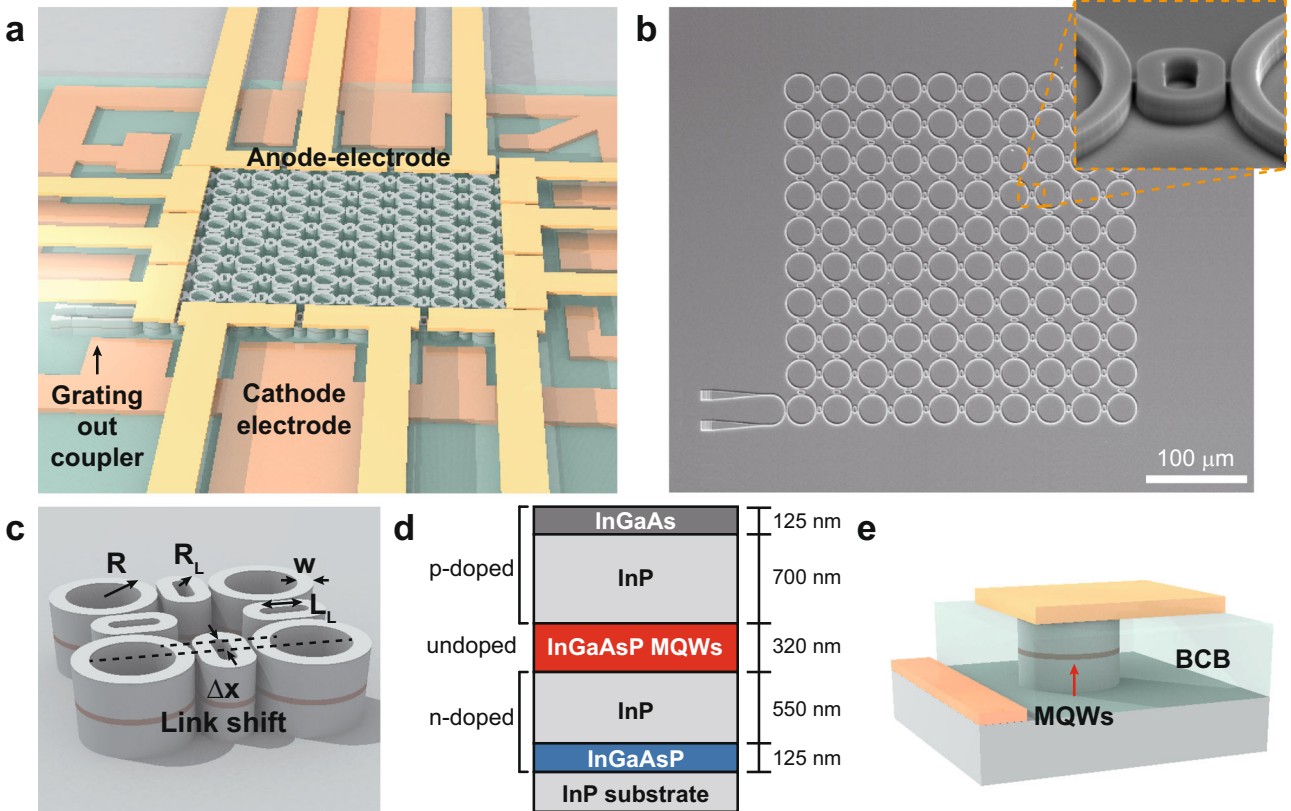

**Fig. 1 Electrically pumped topological insulator laser structure. a** A schematic illustration of an electrically pumped topological insulator laser lattice comprising of a 10 × 10 network of microring cavities that are coupled to each other through link resonators. **b** Scanning electron microscopy (SEM) image of a fabricated topological insulator laser array. The scale bar is 100 μm. **c** A plaquette consisting of four site microrings and four links. **d** A schematic of the epitaxially grown InGaAsP/InP heterostructure wafer structure shows the location of the gain region in the resonators. **e** The position of anode and cathode electrodes with respect to the resonators.

doped InP cladding layers (for details about the wafer structure and fabrication steps see Supplementary Note 1). Considering that the mobility of holes is significantly lower than that of electrons, in our current design, the anode electrodes are positioned directly on top of the cavities, while the cathode electrodes are incorporated at the side of the lattice. Finally, in order to avoid uneven current injection into a large number of edge resonators, we used a set of 12 anode-electrodes (3 on each side of the lattice).

**Characterization.** We first examine the fabricated electrically pumped structures under optical pumping. To characterize the samples, we use a micro- photoluminescence setup to measure the emission spectrum and observe the intensity profile of the lasers. In order to selectively provide optical gain to the topological edge mode, only the outer perimeter of the lattice is optically pumped using a pulsed laser with a nominal wavelength of 1064 nm (15 ns pulse width and 0.4% duty cycle). The pump is applied from the back of the sample (through the substrate) and the emission spectrum is collected from the same side. A set of intensity masks and a knife-edge are used to image the desired pump profile on the sample (The details of the measurement setup can be found in Supplementary Note 4). Figure 2a shows the scattered emission intensity profile of the pumped array. In order to characterize the spectral content of the emitted intensity, we measure the spectrum of the light emitted from the perimeter of the array and the grating output coupler. Figure 2b shows the photoluminescence spectra when the topological cavity is optically pumped at a peak pump intensity of 15.4 kW/cm². A sharp single-mode resonance is

observed from the peripheral sites, as well as the output grating coupler. To further investigate the topological edge mode, the emission spectrum from the output grating was measured as a function of peak pump intensity. As shown in Fig. 2c, the laser remains single moded over a wide range of pump intensities up to ~25 kW/cm². Figure 2d shows the light-light curve associated with this device where the threshold of lasing is 7.84 kW/cm². These results clearly confirm the presence of spectrally single frequency edge modes in this lattice under optical pumping.

Next, we measure the emission properties of the topological laser arrays under electrical pumping. Figure 3a shows a microscope image of the fabricated electrically pumped TIL array used in our study. In this figure, the cathode electrodes (dark orange), vertically separated by a thick layer of benzocyclobutene (BCB), appear in a different color from the anode electrodes (bright orange). A collection of twelve individually controlled anode electrodes ensures that current is uniformly applied to all the edge elements. On the other hand, all cathode branches are connected to each other. To characterize the electroluminescent properties of the topological insulator lasers, a pulsed current driver (ILS Lightwave LDP-3840B) is used (duration: 300 ns, period: 50 μs). The pulsed pumping prevents the detuning of the edge elements from the rest of the array due to heating. Figure 3b shows the collected electroluminescence emission profile from the topological insulator laser. The slightly uneven scattering intensity profile is attributed to the inhomogeneities in the structure since the emission is collected from the back of the sample. In our electrically pumped samples, because of the non-uniformities of the BCB layer, the gratings remained unpumped. As a result, here, instead, the spectra were measured

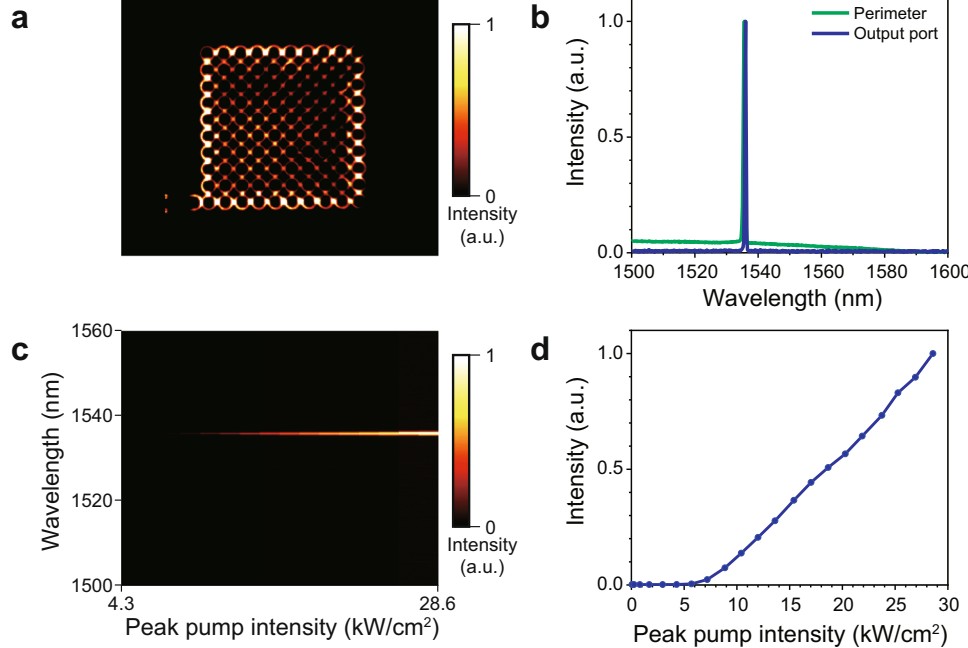

**Fig. 2 Topological edge mode lasing under optical pumping. a** Photoluminescence image from the topological laser array when only its perimeter is pumped. **b** Emission spectra from the grating output coupler and a site on the perimeter at a pump intensity of 15.4 kW/cm². **c** Evolution of the spectrum as a function of the peak pump intensity. The single-mode emission with the narrow linewidth is attributed to the topological edge mode. a.u., arbitrary units. **d** Measured output intensity as a function of the peak pump power. The data presented in **c** and **d** were collected from the output port (grating).

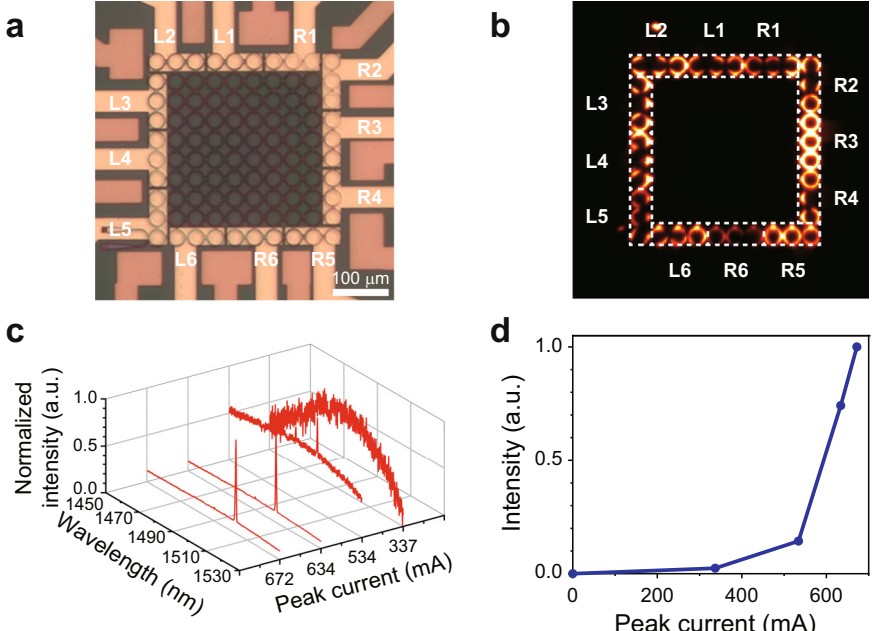

**Fig. 3 Electroluminescence measurements of the topological insulator laser. a** The microscope image of the fabricated topological insulator laser array (top view). In order to selectively excite the topological edge mode, electrodes are designed only at the perimeter of the array (dark orange: cathode, bright orange: anode). R and L labels indicate that the position of the metal pad connected to the electrode is on the right and left, respectively. The scale bar is 100 μm. **b** Intensity profile image of topological insulator laser array when all peripheral sites are pumped at the same level. **c** Spectral evolution of the laser emission as a function of the peak pump current. **d** Measured light-current curve of the laser. The resolution of the spectrometer is 0.2 nm.

from the scattered light emanating from the sites located at the perimeter of the lattice. Figure 3c shows the emission spectrum of the device as a function of the injected peak pulsed current. This spectrum was collected from the brightest peripheral site (R3). The plot shows that as the injected current is increased, a sharp single-mode lasing peak appears along with a rapid increase in

the emitted intensity. It should be noted that the spectra measured at other sites of this lattice exhibit the same emission peak wavelength albeit with different intensities (see Supplementary Note 5). Figure 3d displays the light-current (L-I) curve of this laser, which shows stimulated emission with a threshold peak current of ~500 mA (equivalent to a threshold peak current of

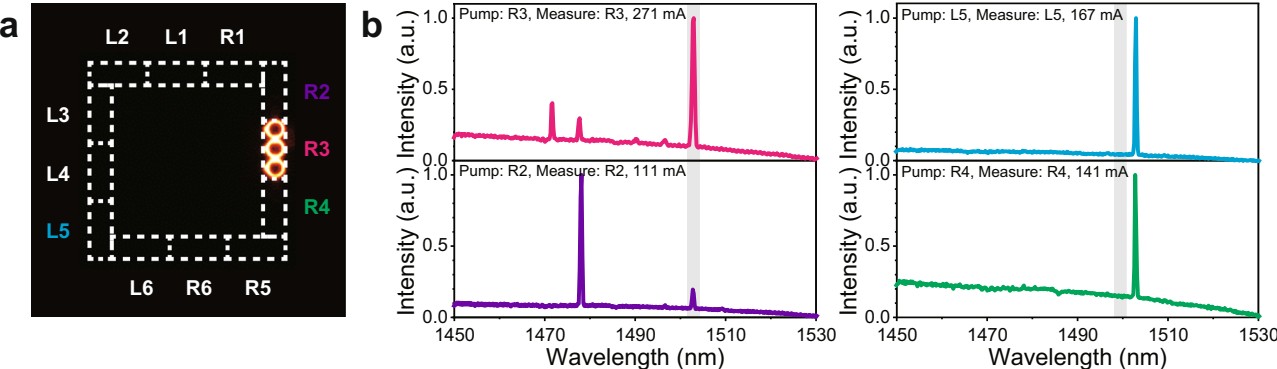

**Fig. 4 Topological edge mode vs. locally excited modes. a** Profile intensity image when the current is injected into only one of the twelve electrodes (in this case R3). **b** Measured electroluminescent spectra when the current is injected into different electrodes. The topological edge mode at a wavelength of 1503 nm persists in all cases (gray shaded area) while in some cases localized modes are excited that prevail the spectrum.

~14 mA per ring or a current density of ~11 kA/cm$^2$). The electrical characteristic curves (I–V curves) of a three-ring laser (when only one electrode is pumped) and the full TIL are provided in the Supplementary Information (please see Note 6). Further experimental observations confirming that topological lasers outperform trivial lattices in terms of spectral purity, as well as robustness to defects, are provided in the Supplementary Information (please see Notes 7 and 8).

## Discussion

In order to verify the nature of the lasing mode and study the relationship between the local modes and the topological edge mode, we change our injection pattern by only applying current to one of the twelve electrodes at a given time (see Fig. 4a). We then measure the emission spectrum from the same site. In this case, we supply significantly larger current levels to be able to assess the presence of the localized modes, as well as the topological edge mode. We also modify the current pulse width to 100 ns to prevent the laser from being damaged due to overheating. Figure 4b shows the measured spectra when various sets of three-ring-resonators (R3, R2, L5, and R4) are pumped. Their spectral response confirms the presence of various locally excited modes in each site. These local modes can belong to individual rings, caused by small but inevitable defects, which can even be formed because of the pump-induced local detunings. While the distribution of the local modes changes from one site to another, the topological edge mode at a wavelength of 1503 nm persists in all these measurements (within the gray shaded area), confirming that it is indeed the collective response from the entire array perimeter elements. Clearly, here the interplay between the edge mode and the pump profile ensures the excitation of the topological edge mode even when a few selected rings at the edge are pumped. When all perimeter elements are coupled to each other, the topological edge mode lases unambiguously while all spurious modes get suppressed. This selective effect that is attributed to the topological nature of modes distinguishes TILs from conventional laser systems. To further confirm that the emission from various sites belongs to an extended topological edge mode, the coherence between various neighboring elements is examined by overlapping their fields and observing the resulting fringes in the image plane (see Supplementary Note 9 for details about coherence measurement). These measurements together with the observed single-mode lasing operation can be an indication of a widespread coherence over the entire edge elements in the topological laser array. However, additional interference measurements are needed between edge elements that are further apart in order to fully validate this claim.

In conclusion, our paper reports the first demonstration of room temperature electrically pumped topological insulator lasers, capable of operating at telecommunication wavelengths. Our topological laser structures are composed of coupled ring resonators and links arranged in an aperiodic lattice and emulate the quantum spin Hall Hamiltonians. The fabrication of these TILs on InGaAsP/InP heterostructures involves precise multi-step lithography/etching/deposition/annealing processes. The lasers clearly show single-frequency emission and the mode appears to be extended across all the gain elements at the periphery of the lattice. Future works may involve more efficient current injection schemes based on suspended graphene sheets as transparent electrodes[51,52]. Furthermore, the chirality (rotation direction) of the topological edge mode can be unambiguously set by incorporating internal S-bends in the micro-resonators(see Supplementary Note 10)[40,53,54]. Our work is expected to pave the way towards the realization of a new class of electrically pumped coherent and phase-locked laser arrays that operate at a single frequency and an extended spatial mode. Such lasers, serving as the primary light source in photonic integrated circuit chips, may have applications in on-chip communications.

## Methods

**Device fabrication.** To fabricate the structures, FOx-16 e-beam resist is spin-coated on a clean piece of wafer and then patterned using high-resolution e-beam lithography followed by development in tetramethylammonium hydroxide (TMAH). The patterns are subsequently transferred into the wafer through a reactive ion etching (RIE) process using H$_2$:CH$_4$:Ar (10:40:7 sccm) plasma. A 240 nm thick silicon nitride (Si$_3$N$_4$) is deposited by plasma-enhanced chemical vapor deposition (PECVD) to ensure insulation and for passivating the sidewalls. To form the metal electrodes, the cathode-electrode area is defined by photolithography. After sequential dry etching of the Si$_3$N$_4$ layer and wet etching of the remaining InP layer, the cathode-electrode is deposited on the InGaAsP n-contact layer using Ni/AuGe/Au (5/75/300 nm). An optically transparent polymer (ben-zocyclobutene, BCB) is spin-coated on the sample for planarization and for separating cathode-electrodes and anode-electrodes at places they overlap. The BCB is etched down using O$_2$:CF$_4$ (10:5 sccm) plasma etching to uncover the top of the ring arrays. The anode-electrodes are defined by photolithography, and then sequential RIE dry-etching and BOE wet-etching processes are performed to remove the Si$_3$N$_4$ layer and FOx-16 resist at those places. The anode-electrodes are formed on the InGaAsP p-contact layer by thermal evaporation of Ti/Au (15/300 nm) and photoresist liftoff. Finally, rapid thermal annealing is performed at a temperature of 380 °C for 30 s and the sample is mounted on a header and wire-bonded for testing. A more detailed description of the fabrication steps can be found in Supplementary Note 1.

**Measurements.** The topological insulator lasers were characterized using a micro-electroluminescence (μ-EL) characterization setup. A pulsed current driver (ILS Lightwave LDP-3840B) is used to supply 300 ns (100 ns, in Fig. 4) current pulse for a period of 50 μs into the topological arrays. Through a set of individual wires connected to each pin of the header, the current is selectively supplied to the desired elements. The light emitted from the back of the sample is collected by a ×10 microscope objective lens with a numerical aperture of 0.26. The emission

from the array is then sent to either a NIR camera (Xenics Inc.) or a spectrometer (Princeton Instruments Acton SP2300) with an attached detector array (Princeton Instruments OMA V). For optical pumping characterizations, a 1064 nm pulsed laser (SPI fiber laser, 15 ns pulses with 0.4% duty cycle) is used as a pumping source. A set of amplitude masks and a knife-edge are used to shape the pump profile and the desired pump distribution is imaged on the sample. Further details of the characterization setup can be found in Supplementary Note 4.

**Reporting summary**. Further information on research design is available in the Nature Research Reporting Summary linked to this article.

## Data availability

The datasets generated during and/or analyzed during this study are available from the corresponding author on reasonable request.

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

## Acknowledgements

We gratefully acknowledge the financial support from DARPA (D18AP00058), Office of Naval Research (N00014-16-1-2640, N00014-18-1-2347, N00014-19-1-2052, N00014-20-1-2522, N00014-20-1-2789), Army Research Office (W911NF-17-1-0481), National Science Foundation (ECCS 1454531, DMR 1420620, ECCS 1757025, CBET 1805200, ECCS 2000538, ECCS 2011171), Air Force Office of Scientific Research (FA9550-14-1-0037, FA9550-20-1-0322), and US–Israel Binational Science Foundation (BSF; 2016381). The authors would like to thank Mordechai Segev from Technion and Patrick Likamwa from CREOL for helpful technical discussions.

## Author contributions

J.-H.C., Y.G.N.L. and W.E.H. designed and fabricated the structures. J.-H.C. and Y.G.N.L. performed the experiments. Simulations were carried out by J.-H.C. and M.P. Finally, J.-H.C., W.E.H., Y.G.N.L., M.P., B.B., D.N.C. and M.K. discussed the results and contributed in preparing the manuscript.

## Competing interests

The authors declare no competing interests.
