## [Peer Review File · Nature Communications]

REVIEWER COMMENTS

Reviewer #1 (Remarks to the Author):

In the manuscript Room temperature electrically pumped topological insulator lasers by Choi et al. the authors experimentally demonstrate a topological laser by using synthetic gauge fields to achieve phase locking, i.e., they rely on the structure which mimics the quantum spin Hall effect for photons. Systems exploiting topological phenomena in photonics are of great interest and have been for a number of years. Topological lasers could become a jewel in the crown of these systems, if they become commercial one day. The results of this paper are a one step towards this achievement. I especially like the fact that synthetic gauge fields were used which in my opinion could lead to new interesting questions and phenomena (an inspiring outlook of using synthetic gauge fields in this context could have been written, but it is not mandatory). Therefore, I would like to recommend this paper for publications in NC.

Reviewer #2 (Remarks to the Author):

This manuscript reports the demonstration of electrically pumped room-temperature topological insulator lasers operating at a telecommunication wavelength. A non-magnetic topological insulator array was fabricated using the InGaAsP/InP wafer structure which was designed for electrical pumping. A single lasing mode was observed at the topological edge when the perimeter of the cavity was either optically or electrically pumped. The authors claimed that topological edge modes were coupled to all perimeter elements, while spurious modes were suppressed.

The manuscript is well written and the technological achievements are impressive. The successful demonstration of an electrically pumped non-magnetic topological insulator laser is of great significance in terms of the practical implementation of a new light source. Therefore, I believe it could be suitable for Nature Communications if the authors could consider the following comments:

1. In the EL measurement of Figure 3, the authors measured the emission spectrum from the brightest region, R3. Please show the measured spectra at other pump sites and check if the lasing wavelengths at these sites are the same as that in R3. In addition, please make the laser image of Figure 3(b) brighter for comparison with the image of Figure 2(a).
2. In Figure 3(d), please state why more points were not measured above threshold. Is there a thermal issue in the above-threshold region? In addition, please state the reason of such a large threshold value (~ 500 mA). Is this larger or similar compared to the lasing threshold under optical pumping?
3. In the structure for electrical pumping (Figure 3), I wonder if the electrical contacts affect the laser performance. Also, please state why no scattered light is observed from the grating out coupler, in contrast to Figure 2. In addition, it would be useful to understand electrical characteristics if the authors show I-V curves.
4. The authors mentioned that chirality can be provided to the topological edge mode by incorporating S-bend in the micro-resonators. I wonder how difficult it is to add S-bend to this design to make unidirectional lasing mode.

Reviewer #3 (Remarks to the Author):

The authors report on the first experimental demonstration of a topological insulator (TI) laser that is both electrically pumped, and operates at room temperature. A TI laser is an array of lasing elements that is made to lase coherently, while being robust to disorder and imperfections. The TI lasers are constructed by introducing gain to the edges of a 2D photonic TI, this in turn causes the edge-states of the array to lase. Since the edge-states are extended, unidirectional and in the bandgap, they tend to not localize in the presence of defects and disorder hence improving lasing properties such as slope efficiency and coherence. For trivial arrays (without a topological bandgap) these imperfections may cause significant decoherence of the lasing elements due to localization effects.

The concept of 2D TI lasers was first introduced in 2018 and was experimentally demonstrated in optically pumped InGaAsP quantum wells in a setup that is geometrically similar to that presented by the authors in the manuscript (Bandres et al, Science 2018). Half a year ago, TI lasers were also demonstrated in a Quantum cascade lasers setup in a more compact way (nano-scale) using the valley Hall effect and electrical pumping in cryogenic temperatures (Zeng et al, Nature 2020). Additionally, TI lasers were demonstrated in optical pumping in nanophotonic topological cavities incorporating III–V semiconductor quantum wells (Smirnova et al, Light Science & Applications 2020)

I found the work to be original and interesting, however as it is written now it only partially supports the claims and conclusions of the authors. The overall novelty of combining electrical pumping and room temperature operation in a micro-scale device without a firm proof of the coherence and efficiency of the device does not meet the criteria of high impact publication in my view, and should be published in a more specialized journal.

Here are my detailed comments:

- 1) **Novelty in concept:** Conceptually, I do not find the manuscript to be particularly novel. The conceptual method for constructing the laser is very similar to [Bandres et al, Science 2018], and the concept of electrical pumping of TI lasers was already demonstrated in a nano scale device in [Zeng et al, Nature 2020] as the authors correctly mentioned. The authors did not explain why changing the pumping mechanism in their platform is a new concept. The new design principles in the manuscript compared to previous works are in the quantum well structure, and in the cathode arrangement, but these seem rather technical or specific, and not general or novel enough. It is true that the authors are the first to show both electrical pumping and room temperature operation together, but as I will explain in the next section, showing the spectrum is not enough in my view to meet all the claims of the authors about a superior device that is robust and coherent.
- 2) **Technical achievement:** since the manuscript is about a practical device, and the claimed novelty regards practical properties (pumping, temperature), it can have high impact if it shows technological relevance or higher performance than previous work.
 - a. **Technological relevance:** The combination of room temperature and electrical pumping is advantageous compared to optical pumping or operation at cryogenic temperatures. However, I think the platform of the authors is still overall less relevant than the nano-scale devices with the valley degree of freedom as in [Zeng et al, Nature 2020], since it is much less compact.

- b.** Proving the topological advantage: The authors focus was on the fact that the lasing spectrum matches the edge-states in the band gap, and I am convinced by the results that indeed the edge states were the lasing modes. However, probing the edge states to lase does not convince me that the lasing properties of the topological structure are superior:
- i.** Although the authors claimed for coherence there is no coherence measurement as in [Smirnova et al, Light Science & Applications 2020], if the authors would show superior coherence it will improve their work greatly.
 - ii.** It will be beneficial if the authors show how their topological device compares to a trivial device. Ex: trivial mode in the topological lattice, or the edge of a trivial version of their lattice. At the current state, there is no way of knowing that the topological structure they have give them an advantage over a trivial structure.
 - iii.** How does the electrical pumping performs compared to the optical pumping, is the transition to electrical pumping problematic for any reason or harmful to the performance?
 - 1.** If it is problematic, how and why is it problematic? what did the authors do to overcome these problem?
 - 2.** If it is not problematic, why is it important to demonstrate it in a proof of concept type of work. Is it surprising that it is not problematic?
 - iv.** The authors claim that their laser is robust to disorder and perturbation, but there is no proof of that by relevant measurements, for example how does the laser preform if a site is missing?
 - v.** The lasing mode should be unidirectional, however the authors did not show this property.

To summarize this section, reading the text did not give me a good notion on how well the topological laser preforms under electrical pumping in room temperature. Furthermore, I am not convinced that all the advantages (robustness, coherence, etc) exist in their platform. Therefore, it is hard to estimate about how promising this method is for future technology.

- 3) In line 139, it says that the cathode electrodes are in red in fig.3a, also the caption directs to red cathodes and orange anodes, but I do not see any red in the figure. Also, the letters R and L are not explained. I think the colors should be changed and the letters explained.
- 4) In fig.3b there is no color map.
- 5) Regarding the results of fig4, to my knowledge, the modes of the IQH are generally not localized, all the modes on the edge are extended, and the modes in the bulk are extended on the bulk. It is claimed that when exciting a small number of sites, localized modes are lasing, my question is where do these localized modes come from? Do they come from disorder on the edge (overcoming the bandgap topology)? Or, are they localized bulk modes? Is the localization a result of the non-homogenous gain, or is it a result of the disorder in the passive system?

- 6) In the supplementary: sometimes it is written (Figure S2) and sometime it is written (Fig. S2) and sometimes there is a space between Fig.S2 (b) and sometime there is no space (Fig.S2(a)).

Reviewer 1

***Reviewer 1.** In the manuscript Room temperature electrically pumped topological insulator lasers by Choi et al. the authors experimentally demonstrate a topological laser by using synthetic gauge fields to achieve phase locking, i.e., they rely on the structure which mimics the quantum spin Hall effect for photons. Systems exploiting topological phenomena in photonics are of great interest and have been for a number of years. Topological lasers could become a jewel in the crown of these systems, if they become commercial one day. The results of this paper are a one step towards this achievement. I especially like the fact that synthetic gauge fields were used which in my opinion could lead to new interesting questions and phenomena (an inspiring outlook of using synthetic gauge fields in this context could have been written, but it is not mandatory). Therefore, I would like to recommend this paper for publications in NC.*

Authors' Response. We sincerely thank our reviewer for the positive assessment of our work and for recommending our manuscript to be published in Nature Communications. As the reviewer noted, our work can potentially bring technologically viable laser arrays based on concepts borrowed from topological photonics one step closer to reality. Per our reviewer suggestion, we have also added a brief overview about synthetic gauge fields in the introductory part of the paper. Please see the main manuscript (Page 2, Paragraph 1, Line 7).

Reviewer 2

Reviewer 2. *This manuscript reports the demonstration of electrically pumped room-temperature topological insulator lasers operating at a telecommunication wavelength. A non-magnetic topological insulator array was fabricated using the InGaAsP/InP wafer structure which was designed for electrical pumping. A single lasing mode was observed at the topological edge when the perimeter of the cavity was either optically or electrically pumped. The authors claimed that topological edge modes were coupled to all perimeter elements, while spurious modes were suppressed. The manuscript is well written and the technological achievements are impressive. The successful demonstration of an electrically pumped non-magnetic topological insulator laser is of great significance in terms of the practical implementation of a new light source. Therefore, I believe it could be suitable for Nature Communications if the authors could consider the following comments:*

Authors' Response. We sincerely thank the reviewer for finding our work of having a great significance and for suggesting that our work is potentially suitable for publication in Nature Communications. Below, we will address the comments raised by the reviewer in the order they appeared in his/her report. When needed, we applied necessary changes in the manuscript and/or the supplementary file. We believe our paper has greatly benefited from the reviewer's excellent technical comments.

Reviewer 2. *In the EL measurement of Figure 3, the authors measured the emission spectrum from the brightest region, R3. Please show the measured spectra at other pump sites and check if the lasing wavelengths at these sites are the same as that in R3.*

Authors' Response. Per our reviewer's suggestion, we have now included the measured spectra at other sites besides R3.

For these data, we electrically pumped all the sites and collected the emission locally by appropriately placing a spatial filter in the image plane. Figure R1 shows the collected emission from sites L2, L4, L5 and R6. In all these cases, the lasing peak was observed at a wavelength of 1503 nm (similar to what was measured at R3). Also, as can be seen in these plots, the spectra all around the cavity exhibit single mode lasing behavior.

Following to this comment, we added a sentence in the main manuscript (Page 7, Paragraph 1, Line 19). Also, we added a new section in the Supplementary file (under Note 5).

Fig. R1 Measured emission spectra under electrical pumping at various sites (L2, L4, L5, and R6).

Reviewer 2. In addition, please make the laser image of Figure 3(b) brighter for comparison with the image of Figure 2(a).

Authors' Response. Per our referee's suggestion, we modified Figure 3 in the main manuscript. Please see Fig. R2 below.

Fig. R2 Intensity profile image of topological insulator laser array when all the peripheral sites are pumped at the same level.

Reviewer 2. In Figure 3(d), please state why more points were not measured above threshold. Is there a thermal issue in the above-threshold region?

Authors' Response. As the reviewer noted, in our current samples, the top electrodes tend to heat up at higher pump levels that makes it difficult to measure the output power at higher pump densities. This could be partially because of the way we performed planarization using BCB. In our future samples, we intend to resolve this issue by using thicker metal electrodes, and by developing new recipes for realizing a more uniform BCB layer across the array.

We have now modified our manuscript to reflect this aspect. Please see the Supplementary Note 1 (Page 2, Paragraph 2, Line 25).

Reviewer 2. *In addition, please state the reason of such a large threshold value (~500 mA). Is this larger or similar compared to the lasing threshold under optical pumping?*

Authors' Response. We'd like to thank the reviewer for bringing up this issue. One should note that our array is composed of 36 ring lasers on the edge. This means that the threshold power for each ring is ~ 14 mA or a current density of ~11 kA/cm². When we fabricated a single ring laser with similar dimensions as each of the rings on the periphery, the threshold current was in the range of ~ 27 mA. As a result, it appears to us that the threshold of the topological array is indeed lower than the threshold required for a collection of the rings (36 × 27 = 972 mA). We observed the same type of response in our optically pumped arrays as well, where the threshold of the topological edge mode appeared to be smaller than the threshold expected from the combination of the same number of single rings. The lower threshold behaviour has been already predicted as one of the features of topological lasers.

To clarify this point, we have now reported the current density in the manuscript. Please see the main manuscript (Page 7, Paragraph 1, Line 20).

Reviewer 2. *In the structure for electrical pumping (Figure 3), I wonder if the electrical contacts affect the laser performance.*

Authors' Response. Yes, in fact, we are still working on the design of the contact layers (in terms of choice of metals and fabrication process) as we also noticed that it is one of the main factors affecting the performance of our lasers. If the reviewer has a suggestion, we will appreciate it.

Reviewer 2. *Also, please state why no scattered light is observed from the grating out coupler, in contrast to Figure 2.*

Authors' Response. In our electrically pumped samples, we were not able to remove the silicon nitride passivation layer (also used to enhance the BCB adhesion) from the top of the gratings because of the non-uniformities of the BCB layer across the wafer. That is why the gratings and the waveguides next to them could not be electrically pumped. Since without pumping, they are highly lossy, they remained dark in the electrically pumped intensity profile images. It should be noted that for gratings to light up, they need to at least be brought to transparency through partial pumping. Since the gratings were not accessible, we instead measured the spectrum through scattered light off the sites.

This point is now clarified in the Supplementary Note 1 (Page 2, Paragraph 2, Line 26). We also added a sentence in the main manuscript (Page 7, Paragraph 1, Line 12).

Reviewer 2. *In addition, it would be useful to understand electrical characterises if the authors show I-V curves:*

Authors' Response. We agree with the reviewer that I-V curves can be useful to understand some of the electrical characteristics of the lasers. We have now added the I-V curves of the topological laser when all the electrodes are connected (Figure R3(a)), and when only one electrode is connected (Figure R3(b)). These results have also been added to the Supplementary file (under Note 6). We also added a sentence in the main manuscript (Page 7, Paragraph 1, Line 23).

Fig. R3 Measured I-V curves from topological insulator lasers when (a) all the electrode are connected, and (b) only one electrode is connected.

Reviewer 2. *The authors mentioned that chirality can be provided to the topological edge mode by incorporating S-bend in the micro-resonators. I wonder how difficult it is to add S-bend to this design to make unidirectional lasing mode.*

Authors' Response. In our previous experiments with optically pumped topological lasers (Bandres *et al*, *Science* **359**, eaar4005 (2018)), we noticed that these arrays tend to exhibit some level of chirality even in the absence of S-bends due to nonlinear hole burning effects. S-bends, however, allowed us to establish chirality in a predetermined fashion. Adding these constructs to our electrically pumped devices, even though possible, requires addressing several additional challenges. For example, the distance between the S-bends and the rings must be precisely controlled. Also, due to the larger width of the waveguides, the tapering of S-bends becomes more difficult. Finally, one should expect much more non-uniformities of BCB layer in the rings with S-bends, thus causing issues with good quality electrodes.

We already designed electrically pumped lasers with S-bends as shown in Fig. R4. At this point, we have only characterized the laser performance under optical pumping (without depositing the electrode). From the strong emission emerging from one side of the grating output coupler, we can confirm that the S-bend design can enforce topological edge mode to become unidirectional. As mentioned above, fabricating uniform top electrodes for both rings and S-bends is quite challenging. We are working on resolving this issue for our future designs.

These results are now added in the Supplementary file (under Note 10).

Fig. R4 (a) Measured intensity profile from a topological insulator array with S-bends inside the ring resonators. (b) A ring resonator with an S-bend. (c) The magnified image of the grating output couplers. The unidirectionality of the topological edge mode can be confirmed from the strong emission observed at only one of the grating couplers.

As our final note, we would like to thank again the reviewer for his/her time. We appreciate the comments we received that helped us to improve the quality of our work. We hope that the reviewer now finds our manuscript suitable for publication in Nature Communications.

Reviewer 3

Reviewer 3. *The authors report on the first experimental demonstration of a topological insulator (TI) laser that is both electrically pumped, and operates at room temperature. A TI laser is an array of lasing elements that is made to lase coherently, while being robust to disorder and imperfections. The TI lasers are constructed by introducing gain to the edges of a 2D photonic TI, this in turn causes the edge-states of the array to lase. Since the edge-states are extended, unidirectional and in the bandgap, they tend to not localize in the presence of defects and disorder hence improving lasing properties such as slope efficiency and coherence. For trivial arrays (without a topological bandgap) these imperfections may cause significant decoherence of the lasing elements due to localization effects.*

The concept of 2D TI lasers was first introduced in 2018 and was experimentally demonstrated in optically pumped InGaAsP quantum wells in a setup that is geometrically similar to that presented by the authors in the manuscript (Bandres et al, Science 2018). Half a year ago, TI lasers were also demonstrated in a Quantum cascade lasers setup in a more compact way (nano-scale) using the valley Hall effect and electrical pumping in cryogenic temperatures (Zeng et al, Nature 2020). Additionally, TI lasers were demonstrated in optical pumping in nanophotonic topological cavities incorporating III–V semiconductor quantum wells (Smirnova et al, Light Science & Applications 2020).

I found the work to be original and interesting, however as it is written now it only partially supports the claims and conclusions of the authors. The overall novelty of combining electrical pumping and room temperature operation in a micro-scale device without a firm proof of the coherence and efficiency of the device does not meet the criteria of high impact publication in my view, and should be published in a more specialized journal.

Authors' Response. We thank the reviewer for finding our work original and interesting. We truly appreciate reviewer's time and effort. In what follows we address the comments made by the reviewer in the order they appear in the report. We hope that the reviewer finds our revised manuscript suitable for publication in Nature Communications.

Reviewer 3. *Novelty in concept: Conceptually, I do not find the manuscript to be particularly novel. The conceptual method for constructing the laser is very similar to [Bandres et al, Science 2018], and the concept of electrical pumping of TI lasers was already demonstrated in a nano scale device in [Zeng et al, Nature 2020] as the authors correctly mentioned. The authors did not explain why changing the pumping mechanism in their platform is a new concept. The new design principles in the manuscript compared to previous works are in the quantum well structure, and in the cathode arrangement, but these seem rather technical or specific, and not general or novel enough. It is true that the authors are the first to show both electrical pumping*

and room temperature operation together, but as I will explain in the next section, showing the spectrum is not enough in my view to meet all the claims of the authors about a superior device that is robust and coherent. Since the manuscript is about a practical device, and the claimed novelty regards practical properties (pumping, temperature), it can have high impact if it shows technological relevance or higher performance than previous work.

Authors' Response. In what follows, we addressed the points brought up by the reviewer regarding the novelty and importance of our work as well as his/her technical questions, in a detailed fashion and with additional experimental results. We hope that our extensive response addresses the reviewer's concerns. We would like to bring it up again to the attention of the referee that our laser operates at room temperature while the only other electrically pumped topological laser to date operates at 9 K (Zeng *et al*, *Nature* **578**, 246 (2020)). This, by all means, marks a major performance improvement for this type of lasers that can motivate future developments in this area.

Reviewer 3. Technological relevance:

The combination of room temperature and electrical pumping is advantageous compared to optical pumping or operation at cryogenic temperatures. However, I think the platform of the authors is still overall less relevant than the nano-scale devices with the valley degree of freedom as in [Zeng et al, Nature 2020], since it is much less compact.

Authors' Response. We respectfully disagree with the reviewer that our platform is less relevant than the other paper published in Nature (Zeng *et al*, *Nature* **578**, 246 (2020)). These are just two different type of designs based on different physics (quantum spin Hall vs. quantum valley Hall), each having their advantages and disadvantages. In terms of being compact, both devices are integrated and compact. In fact, we noticed that by comparing sizes, our laser array is smaller than the one reported by Zeng *et al*. Finally, if the comparison is going to be done based on the ability to scale up power (which is the main purpose for this type of lasers), microring and waveguide type lasers are so far proved to be more suitable for high power applications.

Reviewer 3. Proving the topological advantage:

The authors focus was on the fact that the lasing spectrum matches the edge-states in the band gap, and I am convinced by the results that indeed the edge states were the lasing modes. However, probing the edge states to lase does not convince me that the lasing properties of the topological structure are superior: Although the authors claimed for coherence there is no coherence measurement as in [Smirnova et al, Light Science & Applications 2020], if the authors would show superior coherence it will improve their work greatly.

Authors' Response. Per our reviewer's request and the suggestion by the Editor, we measured the interference patterns formed by the emission from various sites along the periphery of the array in order to confirm that our topological laser elements emit in a coherent fashion. To do so,

one way is to follow the approach used by Smirnova *et al*, where the emission from two sites are superimposed in the far-field. Figure R5 shows the characterization result of the interference from neighbouring rings in our topological array, using this technique. It should be noted that, while this approach works fairly well for very small structures, it is not suitable for larger lasers and when the elements are further away from each other. Since the size of the laser cavity in the paper by Smirnova *et al*, is only ~ 3 times the lasing wavelength, their small topological laser is a perfect candidate for this type of measurements.

Fig. R5 (a) Measured emission intensity profile from electrically pumped topological insulator laser array at image plane and (b) at far-field (Fourier plane).

In order to measure the coherence between those elements of our topological array that are further away from each other, we modified our measurement setup by adding two branches operating as a Mach-Zehnder interferometer. Figure R6 shows a schematic diagram of this modified setup. Each interferometer arm has a moving iris that can select the emission from a desired site. These emissions are then overlapped and the resulting interference fringes are imaged in the camera. Figure R7 shows the measured interference images obtained by overlapping emission from different sites when the perimeter of the topological insulator lasers is electrically pumped. Our experimentally obtained interference fringe patterns clearly show that the emission from these sites are coherent with respect to each other, further confirming our claim that indeed the observed emission belongs to the topological edge mode. It should be noted that we did not observe such interference fringes when examined the emission from trivial arrays.

We have now included these measurements in our manuscript. Please see the main manuscript (Page 8, Paragraph 2, Line 23). We also added the paper by “45. Smirnova, D. *et al*. Room-temperature lasing from nanophotonic topological cavities. *Light Sci Appl* **9**, 127 (2020).” as a new citation in our main manuscript. Also, we added a new section in the Supplementary file (under Note 9).

Fig. R6 Schematic of the micro-electroluminescence characterization setup with the added Mach-Zehnder interferometer for measuring coherence.

Fig. R7 (a) Measured intensity profile of interference fringes when the emission of two different site rings are overlapped. (b) Magnified interference pattern image.

Reviewer 3. It will be beneficial if the authors show how their topological device compares to a trivial device. Ex: trivial mode in the topological lattice, or the edge of a trivial version of their lattice. At the current state, there is no way of knowing that the topological structure they have give them an advantage over a trivial structure.

Authors' Response. To compare the lasing properties of the topological and trivial lattices, we characterized a topologically trivial lattice that has no position shift of the link resonators ($\Delta x =$

0). This corresponds to $\alpha = 0$ case. Unlike the emission spectra collected from the topological insulator laser array, when the perimeter of the trivial lattice is pumped, multiple longitudinal lasing modes appear in the spectrum. Figure R8 compares the lasing spectra of topologically trivial and non-trivial lasers when the perimeters of the arrays are pumped at the same level. Please notice that for demonstration purposes the spectrum of trivial lattice is scaled up by a factor of 3. It should be noted that these measurements were performed on samples of electrically pumped devices before the deposition of the electrodes. After depositing the metal electrodes, the trivial lattice showed no lasing by electrical pumping, whereas the topological array lased as shown before.

To address the reviewers' comment, we have now added a new section in the Supplementary file (under Note 7). We also addressed this point in the main manuscript (Page 8, Paragraph 1, Line 2).

Fig. R8 Measured emission spectra of topologically trivial and non-trivial laser arrays under the same pump level.

Reviewer 3. *How does the electrical pumping performs compared to the optical pumping, is the transition to electrical pumping problematic for any reason or harmful to the performance?*

1. *If it is problematic, how and why is it problematic? what did the authors do to overcome these problem?*
2. *If it is not problematic, why is it important to demonstrate it in a proof of concept type of work. Is it surprising that it is not problematic?*

Authors' Response. We thank the reviewer for raising these questions. We start by answering why it is important to demonstrate a proof of concept electrically pumped topological laser. Topological laser arrays have been introduced to address the issue of power scaling in semiconductor (SC) lasers. While *electrically pumped semiconductor lasers* are one of the most technologically important type of lasers because of their high quantum efficiency, compactness, and integrability, the quality of their emission (spectrally and spatially) tend to drop significantly as their size increases and even as they couple to each other in an array. Topological architecture is proposed as a strategy to increase the radiance (power in single mode) emitted from an array of SC lasers. Our 2018 work confirmed the validity of this claim by showing an optically pumped

topological laser array that outperformed similar but trivial cavities in terms of output power and spectral purity. However, our previous work did not show that this idea is applicable to electrically pumped structures. Our current proof of concept experiments confirm that such a possibility exists. Consequently, those works in the area of high power SC lasers can start adopting this technique. This paper is expected to motivate future works to optimize the performance of the laser through the processes that are already known.

The next question raised by the reviewer is: if and why it is challenging to transition from optically to electrically pumped topological lasers. The answer is yes- it is quite challenging. In fact, the transition from a passive system to an optically pumped structure is rather straightforward since in both cases one can use the same geometry (for example: nanowire high contrast waveguides). This is because in optically pumped lasers, the mechanisms for pumping and mode confinement are rather independent from each other. On the other hand, it is not possible to decouple these two aspects in electrically pumped lasers. In other words, one needs to design and optimize the cavity (both single resonators and the array) and electrical injection mechanisms (wafer structure) together. Electrodes that are typically made from lossy metals add to the complexity of the design as their dissipation loss can turn off the laser altogether. The current injection requires thick cladding layers that severely affect the mode confinement and therefore coupling between adjacent elements. Finally, thermal issues because of excess joule heating at the electrodes and because of leakage, tend to adversely affect the performance of the electrically pumped lasers. Considering all these aspects, it is by no means clear if an optically pumped cavity design can one day be shown to lase under electrical injection. For example, optically pumped spasers (Oulton *et al*, *Nature* **461**, 629 (2009) and Noginov *et al*, *Nature* **460**, 1110 (2009)) has not yet been shown under electrical pumping. Another example is photonic crystal lasers that it took several years to transition from optically (Painter *et al*, *Science* **284**, 1819 (1999)) to electrically pumped devices (Park *et al*, *Science* **305**, 1444 (2004)). It was only after this crucial demonstration that photonic crystal laser research was able to successfully transition to the industry.

Reviewer 3. *The authors claim that their laser is robust to disorder and perturbation, but there is no proof of that by relevant measurements, for example how does the laser perform if a site is missing?*

Authors' Response. To justify our claim on the robustness of our topological insulator lasers to disorder, we examined a topological insulator structure with a site ring missing (un-pumped) on the perimeter element (Fig. R9). When all the perimeter elements of the array are electrically pumped via the twelve electrodes, a single mode lasing peak is observed from different pumping sites. This result shows that our electrically pumped topological insulator lasers exhibit topological protection even if some of site rings are not pumped/missed.

Fig. R9 (a) Intensity profile images of topological insulator laser when two site rings of the topological edge element are not electrically pumped. (b-f) Measured electroluminescent spectra at each different site.

Following to this comment, we now discussed the robustness of our topological insulator laser in the Supplementary file (under Note 8). We also addressed this point in the main manuscript (Page 8, Paragraph 1, Line 2).

Reviewer 3. *The lasing mode should be unidirectional, however the authors did not show this property.*

Authors' Response. We should note that our structure is not made to lase in a predetermined unidirectional fashion (using S-bends). In quantum spin Hall based topological lasers, pumping can excite both CW and CCW modes. While no unidirectionality is expected from passive quantum spin Hall photonic arrangements, the lasers based on this effect tend to show some degree of chirality because of the hole burning mechanism in inhomogenously broadened gain systems. However, since we did not collect the emission from the gratings, we cannot verify if and to what extent the emission is unidirectional. Nevertheless, for practical applications, spontaneous breaking of chirality has no advantage.

We are currently working on adding S-bends to our electrically pumped topological lasers to enforce predetermined unidirectionality in our lasers. However, adding these constructs, even though possible, requires overcoming several additional challenges. For example, the distance between the S-bends and the rings must be precisely controlled. Also, due to the larger width of the waveguides, the tapering of S-bends becomes more difficult. Finally, one should expect much more non-uniformities of BCB layer in the rings with S-bends, thus causing issues with good quality electrodes.

We designed the electrically pumped lasers with S-bends as shown in Fig. R10, and characterized the laser performance under optical pumping (without depositing the electrode). From the strong emission emerging from one of the grating output couplers, we can confirm that the S-bend design leads to unidirectional operation. As mentioned above, fabricating uniform top electrodes for both rings and S-bends is quite challenging. We are currently working to find a solution for this problem for our future designs.

These results are now added in the Supplementary file (under Note 10).

Fig. R10 (a) Measured intensity profile from a topological insulator array with S-bends inside the ring resonators. (b) A ring resonator with an S-bend. (c) The magnified image of the grating output couplers. The unidirectionality of the topological edge mode can be confirmed from the strong emission observed at only one of the grating couplers.

Reviewer 3. *To summarize this section, reading the text did not give me a good notion on how well the topological laser performs under electrical pumping in room temperature. Furthermore, I am not convinced that all the advantages (robustness, coherence, etc) exist in their platform. Therefore, it is hard to estimate about how promising this method is for future technology.*

Authors' Response. We hope that our extensive response and our additional measurement data addresses the concerns of the reviewer.

Reviewer 3. *In line 139, it says that the cathode electrodes are in red in fig.3a, also the caption directs to red cathodes and orange anodes, but I do not see any red in the figure. Also, the letters R and L are not explained. I think the colors should be changed and the letters explained.*

Authors' Response. We revised the main manuscript (Page 7, Paragraph 1, Line 3) and the caption of Figure 3. In addition, we added one sentence to the main manuscript (Page 20, Paragraph 1, Line 4).

Reviewer 3. In fig.3b there is no color map.

Authors' Response. Per reviewer's request, a color map has been added to Figure 3(b).

Fig. R11 Intensity profile image of topological insulator laser array when all peripheral sites are pumped.

Reviewer 3. Regarding the results of fig4, to my knowledge, the modes of the IQH are generally not localized, all the modes on the edge are extended, and the modes in the bulk are extended on the bulk. It is claimed that when exciting a small number of sites, localized modes are lasing, my question is where do these localized modes come from? Do they come from disorder on the edge (overcoming the bandgap topology)? Or, are they localized bulk modes? Is the localization a result of the nonhomogenous gain, or is it a result of the disorder in the passive system?

Authors' Response. As the reviewer stated, topological edge modes are by nature extended (not localized). When all sites in the edge of the topological array are pumped, the topological edge mode will be predominantly excited, which results in the suppression of the local (defect) modes (as we have shown in our measurements). On the other hand, when a single site is pumped (not the entire array), in addition to the extended topological edge mode, it is possible for the local modes to get excited too. This is the situation depicted in Fig. 4. In these cases, the topological edge mode is not yet strong enough to suppress the local modes that are pumped more strongly. The spectra displayed in Fig. 4 is provided to further confirm that the topological edge mode is excited in our system; it overlaps with all the peripheral elements, and it is the reason that the local modes are getting suppressed. We are not certain about the nature of these local modes. They can be modes of the individual rings, caused by small but inevitable defects, our can even be formed because of the pump induced local detunings. We have of course no way to identify their nature experimentally. However, regardless of their origin, their disappearance when the entire array is pumped can only be justified if an extended topological edge mode is getting excited, strongly overlapping with all the rings and completely depleting gain for the local modes.

To clarify this point, we have now added one sentence in the revised manuscript. Please see main manuscript (Page 8, Paragraph 2, Line 14).

Reviewer 3. *In the supplementary: sometimes it is written (Figure S2) and sometime it is written (Fig. S2) and sometimes there is a space between Fig.S2 (b) and sometime there is no space (Fig.S2(a)).*

Authors' Response. These inconsistencies are rectified now.

We would like to thank the reviewer for his/her time. We believe our manuscript has greatly benefited from this review process. We sincerely hope that the reviewer now finds our work suitable for publication in Nature Communications.

REVIEWER COMMENTS

Reviewer #2 (Remarks to the Author):

I think the authors addressed properly my comments/questions in the revised manuscript. I also read the response to reviewer #3 and I believe the authors fully and accurately explained and proved the advantages of their topological insulator lasers. I am happy to recommend publication of this paper as is.

Reviewer #3 (Remarks to the Author):

1. "We respectfully disagree with the reviewer that our platform is less relevant than the other paper published in Nature (Zeng et al, Nature 578, 246 (2020)). These are just two different type of designs based on different physics (quantum spin Hall vs. quantum valley Hall), each having their advantages and disadvantages. In terms of being compact, both devices are integrated and compact. In fact, we noticed that by comparing sizes, our laser array is smaller than the one reported by Zeng et al. Finally, if the comparison is going to be done based on the ability to scale up power (which is the main purpose for this type of lasers), microring and waveguide type lasers are so far proved to be more suitable for high power applications."

The reason I wrote that the Valley-Hall implementation of Zeng et al, Nature 578, 246 (2020) is more compact, is because in the Valley-Hall implementation the array has a similar size as the wavelength of the light which is $\sim 0.1\text{mm}$. While in the ring implementation of the authors, the wavelength is 1500nm but the structure size is much large than their wavelength and of the order of hundreds of microns. Thus, a Valley Hall structure has the potential to support NIR lasers in nano-scale devices, while the ring array will be much harder to compress and therefore is in principle less compact for a given wavelength.

2. Per our reviewer's request and the suggestion by the Editor, we measured the interference patterns formed by the emission from various sites along the periphery of the array in order to confirm that our topological laser elements emit in a coherent fashion. To do so, one way is to follow the approach used by Smirnova et al, where the emission from two sites are superimposed in the far-field...

I appreciate the added data that can indeed improve the manuscript, however the authors should mention which sites they are interfering in the image they show. I could not find in the manuscript, supplementary, or the reply if the sites that are interfered are neighboring sites, or sites that have long distance from each other in the array. This is important in order to estimate the level of coherence. In case it is hard to measure the coherence sites at opposite ends, the authors should at least show the comparison with trivial case (that they mention), so the reader will see the level of improvement in coherence compared to a trivial structure (even if it is for shorter distance than edge to edge). Right now without knowledge of the distance between the interfered sites or the comparison with the trivial image, the measurements of coherence do not add much in my opinion.

3. To compare the lasing properties of the topological and trivial lattices, we characterized a topologically trivial lattice that has no position shift of the link resonators...

Since the spectrum of the trivial laser is clearly less coherent than the topological structure, I am satisfied with this result, since it shows me that the topological laser has improved properties in terms of single mode lasing compared to a trivial lattice.

4. We thank the reviewer for raising these questions. We start by answering why it is important to demonstrate a proof of concept electrically pumped topological laser. Topological laser arrays have been introduced to address the issue of power scaling in semiconductor (SC) lasers...

I suggest that the authors add a sentence in the introduction that summarizes the second part of the reply to my question. Nature communications is intended for a broad audience, and electrically pumped lasers are common in technology. It will be beneficial for the reader if he/she understands the challenges regarding electrical pumping of topological laser described in the reply. i.e the crucial part of finding an electrical structure that fits the special geometry of the topological structures.

5. To justify our claim on the robustness of our topological insulator lasers to disorder, we examined a topological insulator structure with a site ring missing (un-pumped) on the perimeter element Fig. R9)...

The authors answered my question regarding the effect of a disorder.

6. We should note that our structure is not made to lase in a predetermined unidirectional fashion (using S-bends). In quantum spin Hall based topological lasers, pumping can excite both CW and CCW modes...

The authors answered my question regarding unidirectional lasing. I assume the image of the S-ring the authors added is optically pumped. This should be mentioned clearly.

7. We hope that our extensive response and our additional measurement data addresses the concerns of the reviewer...

I am more convinced than previously about the claimed topological advantages of the experimental system of the authors due to the disorder demonstration, and the comparison to the trivial case. I do not find the interference measurements convincing since I could not find the crucial data of what sites are being interfered and the comparison to the trivial case. Therefore, I do not think that the conclusion that these measurements indicate coherence in all sites is accurate, unless more data about the location of the interfered sites is provided.

To conclude, I can recommend the publication of the work, after properly addressing remaining open issues in this reply.

Reviewer #2 (Remarks to the Author):

I think the authors addressed properly my comments/questions in the revised manuscript. I also read the response to reviewer #3 and I believe the authors fully and accurately explained and proved the advantages of their topological insulator lasers. I am happy to recommend publication of this paper as is.

We thank the reviewer for his/her time. We are happy that the reviewer now finds our paper suitable for publication in Nature Communications.

Reviewer #3 (Remarks to the Author):

1. “We respectfully disagree with the reviewer that our platform is less relevant than the other paper published in Nature (Zeng et al, Nature 578, 246 (2020)). These are just two different type of designs based on different physics (quantum spin Hall vs. quantum valley Hall), each having their advantages and disadvantages. In terms of being compact, both devices are integrated and compact. In fact, we noticed that by comparing sizes, our laser array is smaller than the one reported by Zeng et al. Finally, if the comparison is going to be done based on the ability to scale up power (which is the main purpose for this type of lasers), microring and waveguide type lasers are so far proved to be more suitable for high power applications.”.

The reason I wrote that the Valley-Hall implementation of Zeng et al, Nature 578, 246 (2020) is more compact, is because in the Valley-Hall implementation the array has a similar size as the wavelength of the light which is $\sim 0.1\text{mm}$. While in the ring implementation of the authors, the wavelength is 1500nm but the structure size is much large than their wavelength and of the order of hundreds of microns. Thus, a Valley Hall structure has the potential to support NIR lasers in nano-scale devices, while the ring array will be much harder to compress and therefore is in principle less compact for a given wavelength.

The reviewer is absolutely right about size per wavelength that is smaller in the Valley-Hall laser of Zeng et al. However, we are not sure if one can meaningfully compare the size and performance of these two systems, since clearly the Zeng et al’s laser does not even operate at room temperature and is not suitable for generating higher output powers. Also, we are not sure if one can assume that laser size can be scaled down to the NIR regime without any consequences.

More importantly, topological insulator lasers have been proposed for high power coherent emission from an ensemble of smaller lasers. For generating a substantial amount of power, either the optical path length has to be increased or the effective width must be enlarged. In this regard, the ring structure is more advantageous. The reviewer may also

need to consider that the Valley-Hall effect happens at the boundary of two crystals- while the lasing mode occupies only a very thin layer at the interface. On the other hand, the quantum spin Hall effect appears at a boundary of only one structure. It means that the ratio of useful (light-generating) area per entire pattern can be smaller for Valley-Hall lasers.

Finally, the ring structure provides the possibility of edge emitting and coupling of output light through bus waveguides, as well as surface emission through second order gratings. On the other hand, the Valley-Hall design of Zeng et al is only suitable as a surface emitting device and requires additional optics to shape the beam.

In short, we really do not see a noteworthy reason as why Zeng et al.'s Valley-Hall design or other photonic crystal based designs are more relevant for topological insulator lasers. Again, if comparison is made based on their ability to generate coherent emission at higher power, ring-type structures appear to be more relevant.

2. Per our reviewer's request and the suggestion by the Editor, we measured the interference patterns formed by the emission from various sites along the periphery of the array in order to confirm that our topological laser elements emit in a coherent fashion. To do so, one way is to follow the approach used by Smirnova et al, where the emission from two sites are superimposed in the far-field...

I appreciate the added data that can indeed improve the manuscript, however the authors should mention which sites they are interfering in the image they show. I could not find in the manuscript, supplementary, or the reply if the sites that are interfered are neighboring sites, or sites that have long distance from each other in the array. This is important in order to estimate the level of coherence. In case it is hard to measure the coherence sites at opposite ends, the authors should at least show the comparison with trivial case (that they mention), so the reader will see the level of improvement in coherence compared to a trivial structure (even if it is for shorter distance than edge to edge). Right now without knowledge of the distance between the interfered sites or the comparison with the trivial image, the measurements of coherence do not add much in my opinion.

The measurement reported in the supplementary data was performed for two neighboring rings. Since it becomes more difficult to measure the coherence for sites that are further apart, we have measured the coherence between consecutive neighboring elements (1 to 2, 2 to 3, 3 to 4, etc.). By inductive reasoning, if the emission from ring 1 is coherent with respect to the emission from ring 2, and the emission from 2 is coherent with respect to the emission from 3, one can conclude that the emission from 1 is coherent with respect to 3,

and so on. The interference measurement results for elements along one edge are shown in Fig. R1(b-j).

Figure R1. (a) A schematic diagram of the topological laser array showing the location of the overlapped site ring elements. (b-j) Measured interference fringes for various elements along one edge of the topological array. Inset: Magnified interference pattern image.

Finally, per reviewer's suggestion, we also experimentally show the lack of fringe formation between emissions from the two neighboring elements of a trivial laser. Please see Fig. R2 (b) that shows the interference pattern for two neighboring elements in a trivial lattice (no fringes could be identified). We replaced Fig. S13 in the Supplementary file by this new figure that shows also the lack of fringes in the interference pattern of the trivial lattice.

Figure R2. Measured intensity profile of interference fringes when the emission of two different neighboring site rings of the (a) topological laser arrays or (b) trivial arrays are overlapped. Inset: Magnified interference pattern image.

We made these points clear by adding a new figure (Fig. S14) to Supplementary Information, Note 9. Figure S.14 shows the interference pattern for the consecutive elements along one edge of the topological laser array.

3. To compare the lasing properties of the topological and trivial lattices, we characterized a topologically trivial lattice that has no position shift of the link resonators...

Since the spectrum of the trivial laser is clearly less coherent than the topological structure, I am satisfied with this result, since it shows me that the topological laser has improved properties in terms of single mode lasing compared to a trivial lattice.

4. We thank the reviewer for raising these questions. We start by answering why it is important to demonstrate a proof of concept electrically pumped topological laser. Topological laser arrays have been introduced to address the issue of power scaling in semiconductor (SC) lasers...

I suggest that the authors add a sentence in the introduction that summarizes the second part of the reply to my question. Nature communications is intended for a broad audience, and electrically pumped lasers are common in technology. It will be beneficial for the reader if he/she understands the challenges regarding electrical pumping of topological laser described in the reply. i.e the crucial part of finding an electrical structure that fits the special geometry of the topological structures.

We thank the reviewer for his/her suggestion. We have now added this point to the introduction part of the paper. Please see Main manuscript, Page 3, Paragraph 2, Lines 12-13.

5. To justify our claim on the robustness of our topological insulator lasers to disorder, we examined a topological insulator structure with a site ring missing (un-pumped) on the perimeter element Fig. R9)...

The authors answered my question regarding the effect of a disorder.

6. We should note that our structure is not made to lase in a predetermined unidirectional fashion (using S-bends). In quantum spin Hall based topological lasers, pumping

can excite both CW and CCW modes...

The authors answered my question regarding unidirectional lasing. I assume the image of the S-ring the authors added is optically pumped. This should be mentioned clearly.

The image of the rings with an S-bend belongs to a sample fabricated on an electrically pumped wafer before the deposition of top electrodes. The measurement results show the output emission intensity when this sample is pumped optically.

We changed the caption in Fig. S15 in the Supplementary Information, Note 10, to clarify this point.

7. We hope that our extensive response and our additional measurement data addresses the concerns of the reviewer...

I am more convinced than previously about the claimed topological advantages of the experimental system of the authors due to the disorder demonstration, and the comparison to the trivial case. I do not find the interference measurements convincing since I could not find the crucial data of what sites are being interfered and the comparison to the trivial case. Therefore, I do not think that the conclusion that these measurements indicate coherence in all sites is accurate, unless more data about the location of the interfered sites is provided.

To conclude, I can recommend the publication of the work, after properly addressing remaining open issues in this reply.

We sincerely thank the reviewer for his/her time. We believe our paper greatly benefited from this review process. We hope that in light of the above clarifications, the referee now finds our claims justified and our paper suitable for publication in Nature Communications.

REVIEWER COMMENTS

Reviewer #3 (Remarks to the Author):

I do not agree with the inductive reasoning suggested by the authors in the response and in the supplement. While the visibility of the fringes is not zero as in the trivial case, the visibility of the fringes is also far from 1 (it is not calculated from the images but it looks low). Thus, even though each two neighboring sites can have some level of coherence, the coherence can decrease rapidly with distance, as in the case of modes which are local on several sites (and countless other examples) so the logic behind the inductive argument is flawed and should be avoided. Thus, I do not see how one can deduct from these measurements that all sites are coherent with each other as written in line 190. The conclusion I can draw is that the coherence is better than the trivial case, which was already known due to the spectrum measurements, and that there is some level of coherence between neighboring sites as seen in the pictures.

On all other issues, I have no further comments.

Reviewer #3 (Remarks to the Author):

I do not agree with the inductive reasoning suggested by the authors in the response and in the supplement. While the visibility of the fringes is not zero as in the trivial case, the visibility of the fringes is also far from 1 (it is not calculated from the images but it looks low). Thus, even though each two neighboring sites can have some level of coherence, the coherence can decrease rapidly with distance, as in the case of modes which are local on several sites (and countless other examples) so the logic behind the inductive argument is flawed and should be avoided. Thus, I do not see how one can deduct from these measurements that all sites are coherent with each other as written in line 190. The conclusion I can draw is that the coherence is better than the trivial case, which was already known due to the spectrum measurements, and that there is some level of coherence between neighboring sites as seen in the pictures.

On all other issues, I have no further comments.

We thank the reviewer for his/her time. In the interest of time, we removed the claim that all sites emit coherently, and instead mentioned that every two peripheral sites exhibit coherence with respect to each other. We also noted that these measurement results together with the observed single mode lasing operation can be an indication of a widespread coherence over the entire edge elements.

Subsequently, we modified the paragraph in the main text (Page 9, Paragraph 1, Line 2-9) as below:

“To further confirm that the emission from various sites belongs to an extended topological edge mode, the coherence between various neighboring elements is examined by overlapping their fields and observing the resulting fringes in the image plane (see Supplementary Note 9 for details about coherence measurement). These measurements together with the observed single mode lasing operation can be an indication of a widespread coherence over the entire edge elements in the topological laser array. However, additional interference measurements are needed between edge elements that are further apart in order to fully validate this claim.”

And changed the Supplementary Note 9. Please see Page 9, Paragraph 1, Line 12-18.

We believe our paper have benefited from this reviewer's comment. We hope that in consideration of our revised manuscript, the referee now finds our claims justified and our paper suitable for publication in Nature Communications.